# Clinical presentation and management outcomes of pediatric lung abscess: A retrospective cohort study

Lina Alshadfan[1]*, Saleh Abualhaj[2,3], Muna Kilani[4], Hisham Hamdan[1], Diana AlSaify[5], Samia Saber Abu Taleb[6], Mohammed K. Al-raqad[7]

**1** Pediatrics and Pediatric Pulmonology, Department of Pediatrics, Faculty of Medicine, Al Balqa Applied University, Salt, Jordan, **2** General Surgery Department, Faculty of Medicine, Al-Balqa Applied University, Salt, Jordan, **3** General Surgery Department, King Hussein Cancer Center (KHCC), Amman, Jordan, **4** Pediatrics and Pediatric Pulmonology, Department of Pediatrics, Faculty of Medicine, The Hashemite University, Zarqa, Jordan, **5** Prince Hamzah Hospital, Amman, Jordan, **6** Department of pediatrics, Jordanian Ministry of Health, Al Salt Hospital, As-Salt, Jordan, **7** INNOVIA Genetic Health Center, INNOVIA Biobank, Amman, Jordan

* Lina.shadfan@bau.edu.jo

## Abstract

### Background

Pediatric lung abscess is an uncommon but potentially serious complication of pneumonia. Limited data exists on its clinical features, diagnostic workup, and outcomes in otherwise healthy children.

### Objective

To describe the clinical characteristics, management strategies, and outcomes of children diagnosed and treated for lung abscess in Jordan.

### Methods

We conducted a retrospective descriptive study of 23 pediatric patients diagnosed with and managed for lung abscess in Jordan. Data collected included demographic characteristics, clinical presentation, laboratory findings, treatment regimens, and outcomes.

### Results

The mean age was 7.2 years (SD 3.1), with 60.9% male and no underlying chronic diseases in any patient. Systemic symptoms such as fever (91%) and hypoactivity (91%) were common. Cough was reported in 87% of cases, with varying patterns (dry, wet, or both). Imaging confirmed that 56.5% of abscesses were located in the left lung and 43.5% in the right lung. TB was systematically excluded using PPD and PCR testing. All patients received initial broad-spectrum antibiotics; 87% required

**Data availability statement:** The data underlying the findings of this study contain potentially identifiable patient information and cannot be made publicly available due to ethical restrictions. Data are available upon reasonable request from the Institutional Review Board (IRB) at Al-Balqa Applied University (dsr@bau. edu.jo) for researchers who meet the criteria for access to confidential data.

**Funding:** The author(s) received no specific funding for this work.

**Competing interests:** Conflict of Interest The authors declare that the research was conducted in the absence of any commercial or financial relationships that could be construed as a potential conflict of interest. This does not alter our adherence to PLOS ONE policies on sharing data and materials.

escalation to include piperacillin–tazobactam combined with aminoglycoside (either amikacin or gentamicin). The mean duration of antibiotic use was 4.8 weeks. Only one patient (4.3%) underwent surgical intervention, and two patients (8.7%) required ICU care. Follow-up imaging 1–2 weeks after completion of therapy showed full resolution in 91% of cases.

## Conclusions

Pediatric lung abscess in otherwise healthy children presents with significant systemic and respiratory symptoms but responds well to prolonged, escalated antibiotic therapy. Conservative management is effective in the majority of cases, with excellent clinical and radiological outcomes and low surgical intervention rates.

---

## Introduction

Lung abscess is an uncommon but potentially severe complication of pneumonia in the pediatric population [1]. It is defined as a localized collection of pus within the lung parenchyma, resulting in necrosis and cavitation [2]. Although the incidence in children is low compared to adults [3], pediatric lung abscess poses unique diagnostic and therapeutic challenges due to its variable clinical presentation, diverse microbiological causes, and the lack of consensus on optimal management strategies [3,4].

Lung abscess is categorized as primary or secondary depending on underlying conditions. Primary lung abscesses occur in otherwise healthy children without underlying structural lung disease and are most often caused by aspiration pneumonia or bacterial infections with common pathogens like Staphylococcus aureus and Streptococcus pneumoniae [2,5]. In contrast, secondary lung abscesses arise due to predisposing conditions such as congenital pulmonary malformations (e.g., congenital pulmonary airway malformation), immunodeficiency syndromes, or chronic aspiration, which create an environment for persistent infection and necrosis [6–8]. Although rare, with an estimated incidence of approximately 0.7 per 100,000 pediatric admissions per year [1,9], lung abscesses can cause significant morbidity if not recognized and managed promptly [10]. Previous studies have reported that primary lung abscesses account for up to 50–80% of pediatric cases [8,11], while secondary abscesses, though less common, often present greater management challenges and higher risk of complications [12].

Children with primary lung abscesses often present with prolonged fever, respiratory distress, persistent cough, and systemic symptoms despite appropriate initial antibiotic therapy for pneumonia [13,14]. Radiographic imaging, primarily chest X-ray and computed tomography (CT), plays a critical role in confirming the diagnosis and differentiating lung abscess from other conditions such as necrotizing pneumonia or empyema [15,16].

The cornerstone of treatment remains prolonged antibiotic therapy, with surgical or interventional drainage reserved for cases unresponsive to medical management [9].

However, the choice of antibiotic regimens, the timing of escalation, and the indications for surgical intervention can vary widely across institutions [3,17].

Despite improvements in imaging and antibiotic options, evidence regarding the clinical course, treatment patterns, and outcomes of pediatric lung abscess remains limited, particularly in resource-limited settings [18,19]. Understanding local patterns of presentation, antibiotic use, response to therapy, and short-term outcomes is vital to guide clinical practice and optimize care for this uncommon but significant condition.

This multi-center retrospective cohort study aims to describe the clinical characteristics, management strategies, and outcomes of children diagnosed and treated for lung abscess in Jordan. By detailing our experience from a resource-limited setting, we hope to contribute to the limited body of literature and provide insight into real-world practices in the management of pediatric lung abscess.

## Methods

### Study design and setting

This study was conducted as a retrospective observational cohort review at Jordanian ministry of health (MoH) related hospitals, with a dedicated pediatric department. All pediatric patients diagnosed and managed for lung abscess between January 2021 and April 2025 were included.

### Study population

Eligible patients were children aged 0–18 years who were admitted with a final diagnosis of primary lung abscess confirmed by radiological imaging (chest X-ray and/or computed tomography [CT] scan). Patients with incomplete medical records were excluded.

Patients with known immunodeficiency disorders (including HIV infection, primary immunodeficiency, malignancy, or chronic immunosuppressive therapy) were excluded from the study. None of the included patients had documented histories of recurrent severe infections suggestive of underlying immune dysfunction.

### Data collection

Demographic and clinical data were collected from electronic medical records and patient charts using a standardized data extraction form. Collected variables included age, gender, weight, weight centile, history of prior pneumonia, presence of chronic diseases or failure to thrive, duration of symptoms before admission, and antibiotic use before admission.

Clinical presentation data included systemic symptoms (fever, hypoactivity, abdominal pain, vomiting, reduced oral intake, tachycardia, hypotension) and respiratory findings (type of cough, shortness of breath, signs of severity such as retractions, tachypnea, cyanosis, hypoxia, and auscultatory findings). Laboratory parameters (hematologic profile, differential counts, inflammatory markers, kidney function tests, electrolytes, microbiological cultures) and imaging results (chest X-ray and CT findings) were documented. Bronchoalveolar lavage was not routinely performed due to its invasive nature and was reserved for selected severe or non-responding cases. In patients requiring surgical drainage, intraoperative samples were obtained for microbiological analysis when feasible.

Details of management were recorded, including ICU admission, need for oxygen therapy, surgical intervention, antibiotic regimens, duration of antibiotic use, and changes in antibiotics. Follow-up outcomes were assessed based on repeat chest X-ray findings 1–2 weeks after treatment completion.

### Statistical analysis

Descriptive statistics were used to summarize patient characteristics, clinical features, laboratory findings, and treatment details. Categorical variables were presented as frequencies and percentages, while continuous variables were summarized as means and standard deviations (SD). Data analysis was performed using Jamovi software version 2.5.4.

## Ethical considerations

The study was approved by the Institutional Review Board (IRB) of Al-Balqa Applied University (26/3/2/1129), the date of approval was 23/7/2025. As this was a retrospective review, informed consent was waived.

The retrospective data were accessed for research purposes between 10 August 2025 and 7 September 2025. During data extraction, the research team had access to identifiable patient information solely for the purpose of accurate record matching. After data extraction was completed, all data were fully de-identified prior to analysis, and no identifiable information was accessible to the authors thereafter.

## Result

### Patient demographics and baseline characteristics

The study cohort included 23 pediatric patients with a mean age of 7.2 years (SD 3.1; range 2–13 years) and a mean weight of 21.4 kg (SD 8.4; range 11–39 kg). Most patients (82.6%) had a weight within the normal centile range, while 17.4% (n = 4) fell below the 5th centile. The majority were male (60.9%, n = 14) and 39.1% (n = 9) were female. Almost all patients (95.7%) had no previous history of pneumonia, and none had chronic diseases or failure to thrive (FTT).

The mean duration of symptoms prior to admission was 7.6 days (SD 4.7; range 1–16 days). None of the patients were on chronic medications. Prior antibiotic use was reported in 60.9% (n = 14) before hospital admission. Tuberculosis was considered in the differential diagnosis and excluded based on clinical evaluation, radiologic assessment, and microbiologic testing when clinically indicated. A tuberculin skin test (PPD) was performed in 30.4% (n = 7) of patients, and TB PCR (GeneXpert) was performed in 8.7% (n = 2) of cases with clinical suspicion. All results were negative. No patient demonstrated clinical, radiologic, or microbiologic evidence of active tuberculosis.. Regarding the side of lung involvement, 56.5% of the abscesses were in left lung (n = 13, 56.5%) and 43.5% were in right lung (n = 10). (Table 1)

### Presenting features

Systemic manifestations were common among the patients diagnosed with lung abscess. Fever and hypoactivity were the most frequent, each observed in 91.3% (n = 21) of cases. Decreased oral intake was reported in 30.4% (n = 7), while abdominal pain and vomiting were each present in 26.1% (n = 6). Tachycardia was documented in 30.4% (n = 7) of patients. Notably, no cases of hypotension were recorded. (Fig 1a)

Respiratory symptoms were variably reported among the cohort with cough reported in 87.0% (n = 20) of patients. Of these, 10 patients (43.5%) had a dry cough, 6 patients (26.1%) had a wet cough, and 4 patients (17.4%) had features of both dry and wet cough. Shortness of breath (SOB) occurred in 34.8% (n = 8), and hemoptysis was observed in 21.7% (n = 5). (Fig 1b)

Approximately half of the patients (52.2%, n = 12) demonstrated clinical signs of severe respiratory compromise. Among these, retractions and tachypnea were the most frequently observed, each present in 43.5% (n = 10). Respiratory distress was also reported in 43.5% (n = 10). Cyanosis and hypoxia were noted in 34.8% (n = 8) of patients with severity signs. (Fig 1c)

### Laboratory findings

The majority of patients had evidence of a significant inflammatory response. C-reactive protein (CRP) was positive in 91.3% and the erythrocyte sedimentation rate (ESR) was elevated in 87.0% of patients. Anemia was noted in 34.8% of patients, while all patients had normal red blood cell counts. Thrombocytosis was common, with 78.3% of patients having elevated platelet counts. Elevated white blood cell count, and neutrophilia were observed in 91.3% of cases. Lymphopenia and eosinopenia were both observed in 69.6% of patients, whereas elevated monocytes were found in 60.9%. Basophils were low in 91.3% of patients. Most patients had normal renal functions and electrolytes. Elevated creatinine and

**Table 1. Patient demographics and baseline characteristics.**

| Characteristics | Overall (N = 23) |
|---|---|
| Age (year) | |
| Mean (SD) | 7.2 (3.1) |
| Range | 2.0 - 13.0 |
| Weight (kg) | |
| Mean (SD) | 21.4 (8.4) |
| Range | 11.0 - 39.0 |
| Weight Centile | |
| Below 5th | 4 (17.4%) |
| Normal limit | 19 (82.6%) |
| Gender | |
| Female | 9 (39.1%) |
| Male | 14 (60.9%) |
| Previous History of Pneumonia | |
| No | 22 (95.7%) |
| Yes | 1 (4.3%) |
| Chronic Diseases or FTT | |
| No | 23 (100.0%) |
| Duration of symptoms before admission (days) | |
| Mean (SD) | 7.6 (4.7) |
| Range | 1.0 - 16.0 |
| Chronic Medication | |
| No | 23 (100.0%) |
| Antibiotic Use before admission | |
| No | 9 (39.1%) |
| Yes | 14 (60.9%) |
| PPD test | |
| No | 16 (69.6%) |
| Yes | 7 (30.4%) |
| TB test PCR | |
| No | 21 (91.3%) |
| Yes | 2 (8.7%) |
| Side of Lung abscess | |
| Left | 13 (56.5%) |
| Right | 10 (43.5%) |

blood urea nitrogen (BUN) levels were seen in 13.0% and 17.4% of patients, respectively. Sodium and potassium levels were normal in all cases. Immunoglobulin workup was performed in 69.6% of patients and was within normal limits for all tested. Pathogen isolation was low. Sputum cultures were positive for MRSA in only 8.7% of cases, while blood cultures were negative in all patients. (Table 2)

## Antibiotic treatment

The mean overall duration of antibiotic therapy following diagnosis was 4.8 weeks (SD 1.4; range 4–8 weeks). On average, patients required a change in their antibiotic regimen after approximately 6.9 days (SD 3.0; range 1–14 days). All patients initially received a vancomycin plus either a third-generation cephalosporin (ceftriaxone) or a carbapenem

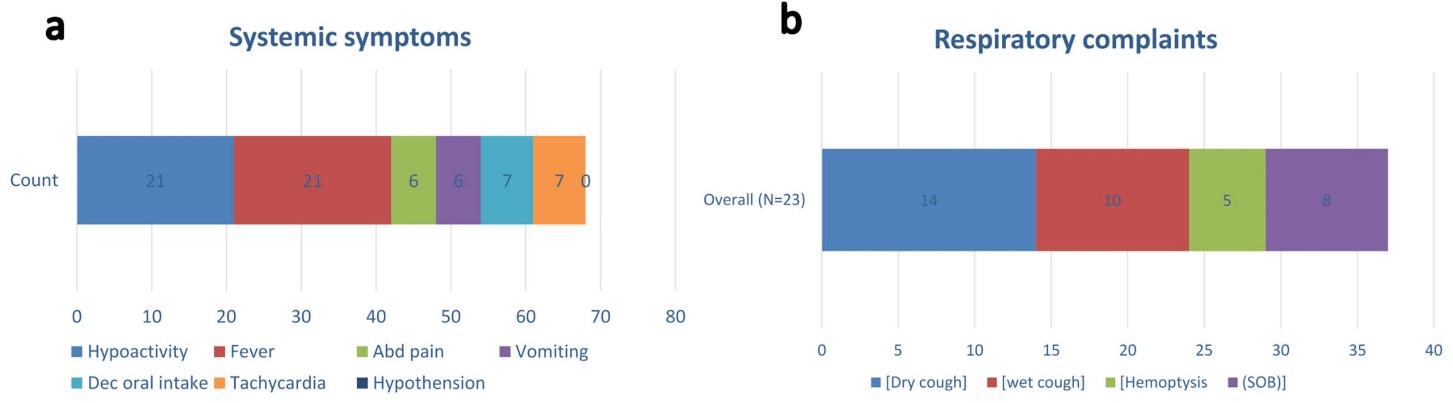

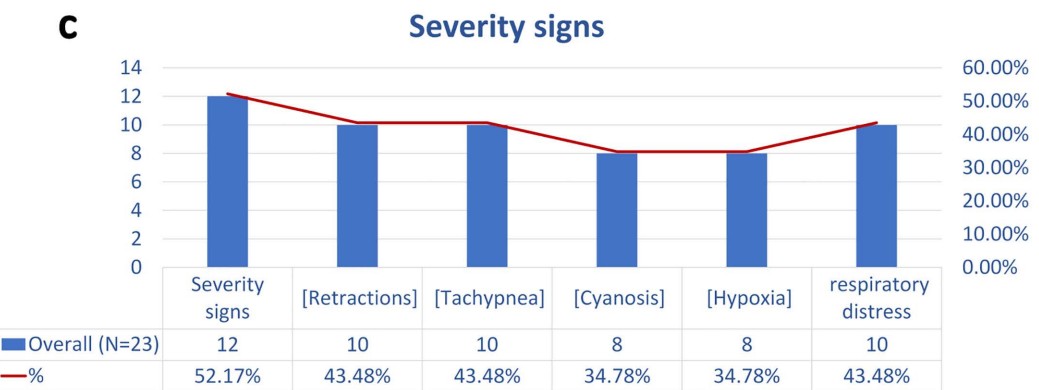

**Fig 1. Clinical presentation and severity indicators among children with lung abscess. a) systemic symptoms, b) Respiratory complaints, and c) Severity signs.**

(meropenem). Patients were monitored closely for 5–7 days to assess their clinical response. If they did not respond adequately to the initial regimen, escalation therapy with aminoglycoside (either amikacin or gentamicin) combined with piperacillin–tazobactam (a beta-lactam/beta-lactamase inhibitor) were used. In this cohort, 60.9% received vancomycin with ceftriaxone, and 26.1% received vancomycin with meropenem. Only 3 patients (13.0%) received Amikacin- Pipera-cillin/tazobactam alone, while the remaining 20 patients (87.0%) required escalation therapy with Amikacin- Piperacillin/tazobactam. Upon discharge, oral amoxicillin–clavulanate only was prescribed in 21.7% of cases, combination of Clindamycin and amoxicillin–clavulanate was administered to 69.6% of patients, and 2 patients received Cefixime, as part of their continued antibiotic therapy. (Table 3)

**Outcome: Hospitalization and Interventions**

The mean duration of hospital admission for the cohort was 16.6 days (SD 4.7; range 10–30 days). Echocardiographic evaluation was performed in some cases, with 78.3% showing no cardiac findings, while 13.0% had normal echocardiography and 8.7% had a minimal pericardial effusion identified. Surgical intervention was rarely required; only one patient (4.3%) underwent video-assisted thoracoscopic surgery (VATS), while the remaining 95.7% were managed non-operatively. Intensive care unit (ICU) admission was necessary for two patients (8.7%), and bronchoscopy was performed in one patient (4.3%). Approximately one-third of patients (34.8%) required oxygen therapy during hospitalization.

**Table 2. Hematologic, inflammatory, and culture results in the cohort.**

| | Overall (N = 23) |
|---|---|
| WBC | |
| High | 21 (91.3%) |
| Normal | 2 (8.7%) |
| Platelet count | |
| High | 18 (78.3%) |
| Normal | 5 (21.7%) |
| RBC | |
| Normal | 23 (100.0%) |
| Neutrophils | |
| High | 21 (91.3%) |
| Normal | 2 (8.7%) |
| Monocytes | |
| High | 14 (60.9%) |
| Low | 2 (8.7%) |
| Normal | 7 (30.4%) |
| Lymphocytes | |
| High | 2 (8.7%) |
| Low | 16 (69.6%) |
| Normal | 5 (21.7%) |
| Eosinophils | |
| High | 2 (8.7%) |
| Low | 16 (69.6%) |
| Normal | 5 (21.7%) |
| Hemoglobin | |
| Low | 8 (34.8%) |
| Normal | 15 (65.2%) |
| Basophils | |
| High | 2 (8.7%) |
| Low | 21 (91.3%) |
| Creatinine | |
| High | 3 (13.0%) |
| Normal | 20 (87.0%) |
| BUN | |
| High | 4 (17.4%) |
| Normal | 19 (82.6%) |
| Na | |
| Normal | 23 (100.0%) |
| K | |
| Normal | 23 (100.0%) |
| CRP | |
| Negative | 2 (8.7%) |
| Positive | 21 (91.3%) |
| ESR | |
| Negative | 3 (13.0%) |
| Positive | 20 (87.0%) |

*(Continued)*

**Table 2.** (Continued)

| | Overall (N = 23) |
|---|---|
| sputum culture | |
| MRSA | 2 (8.7%) |
| negative | 21 (91.3%) |
| Blood Culture | |
| Negative | 23 (100.0%) |

A follow-up chest X-ray was performed in all patients one to two weeks after completing antibiotic treatment; this showed complete resolution in 91.3% of cases, while 8.7% had residual radiological changes. (Table 3)

**Association between age, symptom duration, and severity of presentation**

Figure 2 illustrates the relationship between patients' age and the duration of symptoms prior to admission, stratified by the presence or absence of severity signs and antibiotic use prior to hospitalization. Patients with severity signs (yellow dots) tended to be older and had a longer duration of symptoms before admission compared to those without severity signs (blue dots). The violin plots demonstrate a wider distribution of ages and symptom durations in the severity group, suggesting that delayed presentation and older age may be associated with more severe clinical features in pediatric lung abscess cases. Whereas patients who had received antibiotics before admission tended to have a longer duration of symptoms (median approximately 8–10 days) compared to those who did not (median approximately 4–5 days). Age distribution was similar between groups. (Fig 2)

## Discussion

This case series describes 23 pediatric lung abscess cases in a resource-limited setting, highlighting that most were primary abscesses in previously healthy children with prolonged symptoms before admission and variable severity. Despite limited resources, most patients responded well to IV combination antibiotics (aminoglycosides and beta-lactamase inhibitors) with step-down to oral therapy. Invasive interventions (surgery, bronchoscopy, ICU) were rarely needed, and most had full radiological resolutions.

These findings provide practical insights into the real-world management of pediatric lung abscess in similar healthcare settings and underscore the importance of timely diagnosis, structured antimicrobial therapy, and careful clinical monitoring. The use of broad-spectrum agents in selected severe cases reflects the pragmatic approach often necessary in necrotizing pulmonary infections, where delayed or inadequate treatment may increase morbidity. Nevertheless, these observations should be interpreted cautiously given the small sample size and retrospective design.

Patient characteristics and presentation were consistent with what is typically reported in the literature. The mean age was 7.2 years, with most patients otherwise healthy and no underlying chronic disease or prior history of recurrent pneumonia. This aligns with previous reports indicating that primary lung abscesses in children often occur in otherwise healthy school-aged patients, commonly following a preceding pneumonia that fails to resolve [20–22].

Clinical symptoms showed that nearly all patients presented with systemic inflammatory signs, including fever (91%) and hypoactivity (91%). Cough was a universal feature, present in 87% of patients, with variability in type: dry cough in 10 patients, wet cough in 6, and both in 4. Notably, signs of respiratory severity such as retractions, tachypnea, cyanosis, and hypoxia were observed in about half the cohort, indicating that lung abscess can present with significant respiratory compromise. Auscultatory findings of crackles were common, present in more than three-quarters of patients. These findings are consistent with previous reports, such as the study by Kuhajda et al., which described that early clinical features of lung abscess can mimic pneumonia, with patients commonly presenting with fever, cough (initially non-productive and

**Table 3. Antibiotic treatment patterns, hospital course, and outcomes (N = 23).**

| Treatment Patterns | Overall (N = 23) |
| --- | --- |
| Overall antibiotic use after diagnosis (weeks) | |
| Mean (SD) | 4.8 (1.4) |
| Range | 4.0 - 8.0 |
| After how many days they Change antibiotics | |
| Mean (SD) | 6.9 (3.0) |
| Range | 1.0 - 14.0 |
| IV antibiotics | |
| Vancomycin and Meropenem | 6 (26.09%) |
| Vancomycin and Ceftriaxone | 14 (60.87%) |
| Piperacillin/tazobactam and Amikacin alone | 3 (13.04%) |
| Piperacillin/tazobactam and Amikacin after the initial treatment | 20 (86.96%) |
| Oral Antibiotics | |
| Clindamycin and Amoxicillin-Clavulanate | 16 (69.56%) |
| Amoxicillin-Clavulanate alone | 5 (21.74%) |
| Cefixime | 2 (8.7%) |
| duration of admission | |
| Mean (SD) | 16.6 (4.7) |
| Range | 10.0 - 30.0 |
| Echo Findings | |
| No | 18 (78.3%) |
| Normal | 3 (13.0%) |
| pericardial effusion | 2 (8.7%) |
| surgical intervention | |
| VATS | 1 (4.3%) |
| no intervention | 22 (95.7%) |
| ICU | |
| No | 21 (91.3%) |
| Yes | 2 (8.7%) |
| Bronchoscopy] | |
| No | 22 (95.7%) |
| Yes | 1 (4.3%) |
| Oxygen therapy | |
| No | 15 (65.2%) |
| Yes | 8 (34.8%) |
| Follow up Chest X-ray | |
| Normal | 21 (91.3%) |
| Residual scar | 2 (8.7%) |

later productive), chest pain, dyspnea, and systemic symptoms like weight loss and fatigue. Our data further supports that, in children, distinguishing lung abscess from severe pneumonia based solely on early symptoms can be challenging, emphasizing the importance of imaging and close clinical monitoring to identify progression and complications [8].

Our findings support previous reports describing pediatric lung abscess as rare but significant, with an incidence of about 0.7 per 100,000 admissions annually. For example, a large 23-year retrospective study, our patients were mostly

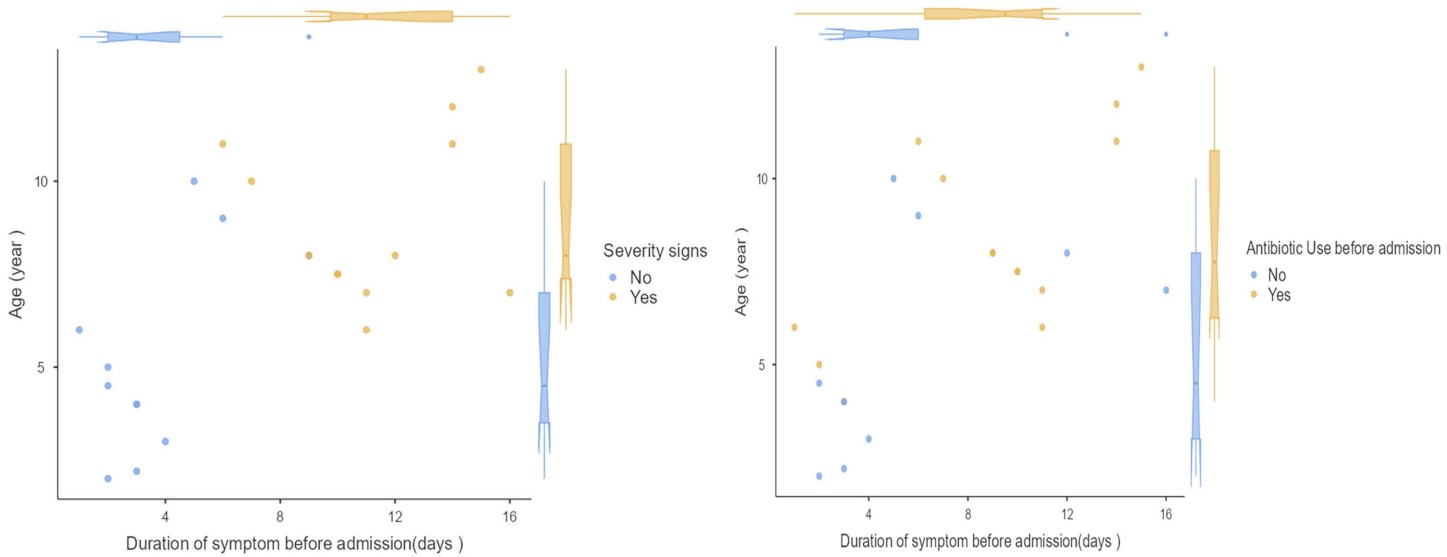

**Fig 2. Relationship between duration of symptoms, age, and key factors (Severity signs and prior antibiotic use) in pediatric lung abscess cases.**

young (mean age 7.2 years vs. 3 years), with fever and cough as the most common symptoms and chest radiography diagnostic in most cases, aided by CT for complex findings. Compared to their cohort's right lung predominance (64%), our cases more often involved the left lung (56.5%). Both studies faced challenges in pathogen identification due to minimally invasive sampling. Management was similarly conservative, with prolonged IV antibiotics (mean 4.8 weeks) and rare need for surgery. Complication rates were low in both, underscoring that conservative treatment is generally effective, even in resource-limited settings [9].

In our cohort, the laboratory profile was largely consistent with published findings on pediatric lung abscesses. As reported by Chirteş et al. [1], severe inflammatory syndromes are common, with significantly elevated inflammatory markers such as CRP and ESR, pronounced leukocytosis with neutrophilia, and evidence of anemia and thrombocytosis. Similarly, our patients showed high inflammatory markers, with 91.3% having elevated CRP and 87% elevated ESR levels. Leukocytosis was seen in 91.3% of cases, primarily driven by increased neutrophil counts (91.3%), while lymphopenia and eosinopenia were also common (69.6% for each), paralleling the left shift described by Kamangar [23].

Like previous reports recommending microbiologic testing for causative organisms, our approach included blood and sputum cultures in all cases, though all blood cultures remained negative, mirroring findings in both Chirteş' case and larger retrospective series [1,23]. PPD and TB PCR tests were performed in selected cases to rule out tuberculosis, in line with the diagnostic approach outlined by Kamangar [23], but no patients were found to have active TB infection.

Hematologic profiles in our patients showed mild anemia in approximately one-third (34.8%), which is consistent with the hypochromic, microcytic anemia observed in other pediatric reports [2,8]. Thrombocytosis was common in comparable studies [24], and in our cohort, platelet counts were elevated in 78.3% of patients, again highlighting the robust inflammatory response characteristic of lung abscesses.

Taken together, these laboratory trends reinforce that pediatric lung abscesses typically present with a marked inflammatory response, prominent neutrophilia, and variable anemia and thrombocytosis — underscoring the importance of comprehensive hematologic and microbiologic workup to guide appropriate treatment while excluding alternative diagnoses such as tuberculosis or parasitic infections when clinically indicated.

Our cohort's management aligns with recommendations that emphasize conservative treatment for pediatric lung abscess, which can achieve up to 90% success with extended antibiotic therapy alone. In the literature, intravenous antibiotics are typically given for 2–3 weeks, followed by oral therapy, totaling 3–6 weeks, with broad-spectrum coverage targeting Gram-positive, Gram-negative, and anaerobic organisms [3,25]. Similarly, all our patients received initial broad-spectrum therapy with vancomycin plus either a third-generation cephalosporin (ceftriaxone) or a carbapenem (meropenem), and most (87%) required escalation with aminoglycoside plus piperacillin–tazobactam after 5–7 days due to insufficient response, reflecting guidance to reassess if no improvement occurs within 7–10 days [25].

Importantly, the choice of broad-spectrum antibiotics in our setting was influenced by several local factors, including the clinical features and severity at presentation, the fact that pneumococcal vaccination is not part of our national immunization schedule, and the limited availability of reliable culture and sensitivity results. Additionally, many patients had already received oral antibiotics prior to admission, potentially affecting initial microbial susceptibility and response to treatment. These factors align with observations from other studies in similar contexts, where incomplete immunization, prior antibiotic exposure, and limited microbiological testing capacity have all been cited as significant challenges in the effective management of pediatric lung abscess [3,8,19,26]. Despite these challenges, only one patient required surgery and two needed ICU care, confirming that timely escalation and prolonged medical management can achieve favorable outcomes, consistent with previous reports [25].

The mean length of stay was 16.6 days, which is comparable to published data on pediatric lung abscesses treated conservatively [9,27]. Importantly, nearly all patients (91%) had complete radiological resolution on follow-up chest X-ray performed 1–2 weeks after antibiotic completion, demonstrating favorable outcomes with appropriate therapy. This finding aligns with the existing evidence that the vast majority of lung abscesses (approximately 85–90%) will respond to systemic antibiotic treatment, with symptoms subsiding within 2–4 weeks and radiographic resolution typically following thereafter [28].

This study provides valuable, context-specific insight into the diagnosis, clinical course, management strategies, and outcomes of pediatric lung abscess in a resource-limited setting. By compiling real-world, multi-center data on a relatively rare condition, it adds valuable evidence to an area where pediatric-specific research remains scarce. The study also highlights practical decision-making in antibiotic selection, diagnostic imaging, and follow-up care, offering observations that may be relevant for similar settings globally. However, its retrospective design inherently limits control over data completeness and standardization of clinical assessments. The small sample size may reduce the generalizability of the findings and may not capture the full spectrum of microbiologic etiologies, rare complications, or less common management approaches such as percutaneous drainage. Furthermore, the lack of routine advanced microbiologic or molecular testing could underestimate pathogen diversity. Larger prospective studies are needed to validate these findings and guide future pediatric lung abscess management.

## Conclusion

Our findings emphasize that pediatric lung abscess, though uncommon, requires a high index of suspicion, systematic microbiological exclusion (especially for TB), and often prolonged broad-spectrum antibiotic treatment. Conservative management is effective in the vast majority of cases, with excellent radiologic outcomes and minimal need for surgical intervention. Future prospective studies may help clarify optimal antibiotic duration, escalation strategies, and indications for invasive management in this population.

## Author contributions

**Conceptualization:** Lina Alshadfan, Saleh Abualhaj, Muna Kilani, Samia Saber Abu Taleb, Mohammed K. Al-raqad.
**Data curation:** Lina Alshadfan, Diana AlSaify.

**Formal analysis:** Saleh Abualhaj, Samia Saber Abu Taleb.

**Investigation:** Lina Alshadfan, Saleh Abualhaj, Muna Kilani, Samia Saber Abu Taleb.

**Methodology:** Lina Alshadfan, Saleh Abualhaj, Diana AlSaify.

**Project administration:** Lina Alshadfan, Hisham Hamdan, Mohammed K. Al-raqad.

**Resources:** Hisham Hamdan.

**Supervision:** Lina Alshadfan, Mohammed K. Al-raqad.

**Validation:** Muna Kilani, Samia Saber Abu Taleb.

**Visualization:** Hisham Hamdan.

**Writing – original draft:** Lina Alshadfan, Saleh Abualhaj.

**Writing – review & editing:** Muna Kilani, Hisham Hamdan, Diana AlSaify, Samia Saber Abu Taleb, Mohammed K. Al-raqad.

## Acknowledgments

We would like to extend our heartfelt gratitude to all individuals who have contributed to the successful completion of this research endeavor. We express our deepest appreciation to the Institutional Review Board (IRB) at Al-Balqa' Applied University for their meticulous review and invaluable feedback, ensuring the ethical conduct of the study and safeguarding the safety of our participants.

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
