## [Decision Letter · Decision Letter 0]

4 Feb 2026

Dear Dr. Alshadfan,

Thank you for submitting your manuscript to PLOS ONE. After careful consideration, we feel that it has merit but does not fully meet PLOS ONE’s publication criteria as it currently stands. Therefore, we invite you to submit a revised version of the manuscript that addresses the points raised during the review process.

We look forward to receiving your revised manuscript.

Kind regards,

Imtiaz Wani

Academic Editor

PLOS One

Journal Requirements:

“Conflict of Interest

The authors declare that the research was conducted in the absence of any commercial or financial relationships that could be construed as a potential conflict of interest”

3. In the online submission form you indicate that your data is not available for proprietary reasons and have provided a contact point for accessing this data. Please note that your current contact point is a co-author on this manuscript. According to our Data Policy, the contact point must not be an author on the manuscript and must be an institutional contact, ideally not an individual. Please revise your data statement to a non-author institutional point of contact, such as a data access or ethics committee, and send this to us via return email. Please also include contact information for the third party organization, and please include the full citation of where the data can be found.

Additional Editor Comments:

There is no p value in the study, would have gained more clinical significance

Reviewers' comments:

Reviewer's Responses to Questions

**Comments to the Author**

1. Is the manuscript technically sound, and do the data support the conclusions?

Reviewer #1: Partly

2. Has the statistical analysis been performed appropriately and rigorously?

Reviewer #1: N/A

3. Have the authors made all data underlying the findings in their manuscript fully available?

Reviewer #1: Yes

4. Is the manuscript presented in an intelligible fashion and written in standard English?

Reviewer #1: Yes

Reviewer #1: It is needed clarification of patient's management:

- It was excluded imunosupression/ HIV? Any child have repetition infections?

- Why some cases started on Meropenem, if there is no microbiologic confirmation or evidence that in some cases there is multidrug resistant bacteria

- There are no place for microbiologic complementary exams? Broncoalveolar lavage/ or intra-operative samples, for those in need for surgery

- Exclusion of TB should be revised, as PPD is done for exclusion of latent TB. Is it possible to perform direct exam or culture?

Minor revisions:

- Replace Tazocin for Piperacilin/tazobactam

- Tables should be simplified

**Do you want your identity to be public for this peer review?** For information about this choice, including consent withdrawal, please see our Privacy Policy

Reviewer #1: No

---

## [Author Response · Author response to Decision Letter 1]

28 Feb 2026

RESPONSE TO REVIEWER #1

We thank the reviewer for the careful evaluation of our manuscript and for the constructive suggestions. We have revised the manuscript to clarify patient selection criteria, microbiologic evaluation, antimicrobial decision-making, and tuberculosis exclusion methods.

1. Was immunosuppression/HIV excluded? Any children with recurrent infections?

Response:

We appreciate this important point. Patients with known immunodeficiency disorders, including HIV infection, primary immunodeficiency, malignancy, or chronic immunosuppressive therapy, were excluded from the study. Additionally, none of the included patients had documented histories suggestive of recurrent severe infections or underlying immune compromise. This clarification has now been added to the Methods section.

Manuscript Modification: Method/Study Population

“Patients with known immunodeficiency disorders (including HIV infection, primary immunodeficiency, malignancy, or chronic immunosuppressive therapy) were excluded from the study. None of the included patients had documented histories of recurrent severe infections suggestive of underlying immune dysfunction.”

2. Why were some cases started on Meropenem without microbiologic confirmation of MDR bacteria?

Response:

We thank the reviewer for this important comment. In our institution, all patients with pediatric lung abscess were managed according to a standardized empiric protocol for severe necrotizing pulmonary infections. Initial therapy consisted of vancomycin combined with either a third-generation cephalosporin (ceftriaxone) or a carbapenem (meropenem), depending on clinical severity and extent of radiologic involvement.

Meropenem was selected in cases presenting with extensive necrosis, systemic toxicity, or concern for resistant Gram-negative pathogens, rather than based on confirmed multidrug-resistant organisms. Given the high rate of culture negativity in our cohort, antibiotic selection was guided by clinical severity and institutional protocols rather than microbiologic confirmation.

Manuscript Modification: Discussion

“This case series describes 23 pediatric lung abscess cases in a resource-limited setting, highlighting that most were primary abscesses in previously healthy children with prolonged symptoms before admission and variable severity. Despite limited resources, most patients responded well to IV combination antibiotics (aminoglycosides and beta-lactamase inhibitors) with step-down to oral therapy. Invasive interventions (surgery, bronchoscopy, ICU) were rarely needed, and most had full radiological resolutions.

These findings provide practical insights into the real-world management of pediatric lung abscess in similar healthcare settings and underscore the importance of timely diagnosis, structured antimicrobial therapy, and careful clinical monitoring. The use of broad-spectrum agents in selected severe cases reflects the pragmatic approach often necessary in necrotizing pulmonary infections, where delayed or inadequate treatment may increase morbidity. Nevertheless, these observations should be interpreted cautiously given the small sample size and retrospective design.”

3. Why were there no microbiologic complementary exams (BAL or intraoperative samples)?

Response:

We thank the reviewer for raising this important point. Bronchoalveolar lavage (BAL) was not routinely performed due to its invasive nature and because most patients were clinically stable and responded to medical therapy. Only one patient in the cohort required surgical intervention; in that case, intraoperative sampling was obtained for microbiological analysis. However, microbiologic yield remained low, likely influenced by prior antibiotic exposure. These clarifications have been incorporated into the Methods section.

Manuscript Modification: Methods:

“Bronchoalveolar lavage was not routinely performed due to its invasive nature and was reserved for selected severe or non-responding cases. In patients requiring surgical drainage, intraoperative samples were obtained for microbiological analysis when feasible.”

4. TB exclusion – PPD is for latent TB; was direct exam/culture done?

Response:

We thank the reviewer for this important clarification. Tuberculosis was excluded based on clinical assessment, radiological findings, and microbiologic testing when clinically indicated. A PPD test was performed in 30.4% (n=7) and TB PCR (GeneXpert) in 8.7% (n=2) of patients who had clinical or radiologic features warranting evaluation. In these cases, results were negative. No patient had clinical, radiologic, or microbiologic evidence suggestive of active tuberculosis. We have clarified this in the revised manuscript.

Manuscript Modification: Result/ Patient Demographics and Baseline Characteristics:

“Tuberculosis was considered in the differential diagnosis and excluded based on clinical evaluation, radiologic assessment, and microbiologic testing when clinically indicated. A tuberculin skin test (PPD) was performed in 30.4% (n=7) of patients, and TB PCR (GeneXpert) was performed in 8.7% (n=2) of cases with clinical suspicion. All results were negative. No patient demonstrated clinical, radiologic, or microbiologic evidence of active tuberculosis.”

Minor Revisions

- Replace “Tazocin”

Done

- Tables should be simplified

We have revised and simplified the tables to improve clarity and readability.

---

## [Editor Report · Decision Letter 1]

3 Mar 2026

Clinical Presentation and Management Outcomes of Pediatric Lung Abscess: A Retrospective Cohort Study

PONE-D-25-61323R1

Dear Dr. Alshadfan,

We’re pleased to inform you that your manuscript has been judged scientifically suitable for publication and will be formally accepted for publication once it meets all outstanding technical requirements.

Kind regards,

Imtiaz Wani

Academic Editor

PLOS One
---

## [Editor Report · Acceptance letter]

PONE-D-25-61323R1

PLOS One

Dear Dr. Alshadfan,

I'm pleased to inform you that your manuscript has been deemed suitable for publication in PLOS One. Congratulations! Your manuscript is now being handed over to our production team.

Kind regards,

on behalf of

Dr. Imtiaz Wani

Academic Editor

PLOS One